# Dyslipidaemia—Genotype Interactions with Nutrient Intake and Cerebro-Cardiovascular Disease

**DOI:** 10.3390/biomedicines10071615

**Published:** 2022-07-06

**Authors:** Sung-Bum Lee, Ja-Eun Choi, Byoungjin Park, Mi-Yeon Cha, Kyung-Won Hong, Dong-Hyuk Jung

**Affiliations:** 1Department of Family Medicine, Soonchunhyang University Bucheon Hospital, Bucheon 22972, Korea; sblee@schmc.ac.kr; 2Department of Healthcare, Theragen Bio Co., Ltd., Daewangpangyo-ro 700, Seongnam-si 13488, Korea; jaeun.choi@theragenbio.com (J.-E.C.); miyeon.cha@theragenbio.com (M.-Y.C.); 3Department of Family Medicine, Yonsei University College of Medicine, Seoul 03722, Korea; bjpark96@yuhs.ac

**Keywords:** nutrients, single-nucleotide polymorphism, dyslipidaemia, cerebro-cardiovascular disease

## Abstract

A comprehensive understanding of gene-diet interactions is necessary to establish proper dietary guidelines to prevent and manage cardio-cerebrovascular disease (CCD). We investigated the role of genetic variants associated with dyslipidaemia (DL) and their interactions with macro-nutrients for cardiovascular disease using a large-scale genome-wide association study of Korean adults. A total of 58,701 participants from a Korean genome and epidemiology study were included. Their dietary intake was assessed using a food frequency questionnaire. Dyslipidaemia was defined as total cholesterol (TCHL) ≥ 240 mg/dL, high-density lipoprotein (HDL) < 40 mg/dL, low-density lipoprotein (LDL) ≥ 160 mg/dL, triglycerides (TG) ≥ 200 mg/dL, or dyslipidaemia history. Their nutrient intake was classified as follows: protein intake: high ≥ 30%, 30% > moderate ≥ 20%, and 20% > low in daily total energy intake (TEI); carbohydrate intake: high ≥ 60%, 60% > moderate ≥ 50%, and 50% > low; fat intake: high ≥ 40%, 40% > moderate ≥ 30%, and 30% > low. Odds ratios and 95% confidence intervals were calculated after adjusting for age; sex; body mass index (BMI); exercise status; smoking status; alcohol intake; principal component 1 (PC1); principal component 2 (PC2); and intake of carbohydrates, fats, and proteins. This analysis included 20,596 patients with dyslipidaemia and 1027 CCD patients. We found that rs2070895 related to *LIPC* was associated with HDL-cholesterol. Patients with the minor allele (A) in rs2070895 had a lower risk of CCD than those carrying the reference allele (G) (odds ratio [OR] = 0.8956, *p*-value = 1.78 × 10^−2^). Furthermore, individuals consuming protein below 20% TEI with the *LIPC* reference allele had a higher risk of CCD than those with the minor allele (interaction *p*-value 6.12 × 10^−3^). Our findings suggest that the interactions of specific polymorphisms associated with dyslipidaemia and nutrients intake can influence CCD.

## 1. Introduction

Cerebro-cardiovascular disease (CCD) is a world-wide common cause of mortality, and its prevalence is increasing in both developed and developing countries [1]. According to a World Health Organization (WHO) report, 17.9 million people died from heart disease and stroke in 2016, accounting for 31% of global deaths, and an estimated 23.6 million people will die from CCD by 2030 [2]. Socio-economic burdens have affected the lifestyles of Koreans. Cardiovascular disease (CVD) is one of the leading causes of mortality in Korea, representing 20% of all deaths in Korea [3]. Therefore, it is critical for individuals to prevent CCD risk factors by modifying lifestyles.

Epidemiological evidence has demonstrated that CCD is related to age; sex; lifestyle risk factors such as unhealthful diet, smoking, alcohol uptake, and low physical activity; and several metabolic diseases such as hypertension, diabetes mellitus (DM), and dyslipidaemia (DL) [4]. Dyslipidaemia can induce many clinical consequences. It increases the deposition of lipids in the arteries, thereby narrowing the lumen of the vessel, resulting in thrombotic events, CAD, stroke, CCD, and pulmonary embolism [5,6]. DL is the most common cause of CCD; the frequency of DL is 87% in Asian patients with CCD and 49% of the patients in the INTERHEART study with a first MI had underlying DL [7]. There are various contributors to CCD and dyslipidaemia. Lipid oxidation is an essential contributor to atherosclerosis leading to CCD, and high levels of LDL and TG and a low level of HDL are associated with coronary artery disease (CAD) and ischemic stroke (IS) in DL [8]. However, despite DL patients maintaining optimal lipid profiles with antidyslipidaemia therapy, CCD remains prevalent in individuals with DL. Miller et al. found that high-dose statin therapy decreased the incidence of CCD; however, patients treated with statins remain at high residual risk for future CCD events [9]. This implies that other irreversible risk factors, such as genetic factors, need to be investigated.

Genetic factors can influence the risk of CCD. Previous genome-wide association studies (GWASs) revealed the genetic susceptibility of CCD [10]. Furthermore, previous studies found the SNPs to be genetically associated with various diseases such as obesity, blood pressure, lipids, DM, CAD, and IS [11]. Nevertheless, the majority of GWASs for CCDs has been conducted in European populations, and few studies have been conducted in Asian populations [12].

Lifestyle factors such as diet are associated with the risk of CCD [13]. Higher diet quality scores, assessed using the Mediterranean-style Diet Score (MDS), Dietary Approaches to Stop Hypertension (DASH) diet score, or the Alternate Healthy Eating Index (AHEI), were associated with lower incidence rates of CAD and IS [14,15,16]. Few studies have examined the genetic associations between nutrients (carbohydrates, proteins, and fats) and CCD.

Here, we conducted GWASs between case–controls of three CVDs (IS, CAD, and CCD) in DL patients and analysed the genetic interactions with nutrients and CVDs according to the proportions of carbohydrates, proteins, and fats using data from the nationwide Korean Genome and Epidemiology Study (KoGES). Our study aimed to identify SNPs that are genetically associated with lipid traits and CVDs in KoGES data. We further aimed to find the genetic pathways linking nutrients and CVDs in the dyslipidaemia group.

## 2. Materials and Methods

### 2.1. Study Population

Our study used the KoGES dataset obtained from the Korean Center for Disease Control and Prevention. The cohort of KoGES was the health examination group (KoGES HEXA), and the dataset consists of the pharmacologic history, anthopomethric traits, and blood biochemistry of participants [17]. The detailed cohort information is described in our previous report [18]. Briefly, KoGES HEXA includes 58,701 participants whose genome-wide SNP data were obtained. We present the exclusion criteria as schematically illustrated in Figure 1. We excluded participants with missing values, i.e., smoking, alcohol, exercise history, and body mass index (BMI) (*n* = 471). Additionally, participants with a malignancy history or no response regarding malignancy were excluded (*n* = 2202). After those exclusions, 56,028 participants were included. Among the filtered subjects, 20,788 had dyslipidemia. Subjects without blood pressure data (*n* = 18) or nutrient intake data (*n* = 174) were excluded. The final sample size for the present analysis was 20,596, and these samples were subdivided by CCD (yes or no), CAD (yes or no), and IS (yes or no). A flowchart of the patient selection process is shown in Figure 1.

### 2.2. Study Design

Our study was performed in four steps (Figure 2). The first step was the preliminary study to choose the SNPs for lipid traits (TCHL, HDL, LDL, and TG) using the HEXA cohort. The second step was the selection of final candidate SNPs by matching the reported SNPs and the replicated SNPs in other Korean cohorts (Ansung, Ansa, Nongchon); this involved selecting the SNPs with the same trend of association between lipid traits and other cohort SNPs (Appendix A). The third step was an analysis of the genetic associations of the intake of macronutrients (carbohydrates, proteins, and fats) and CVDs in the dyslipidaemia patients of the HEXA cohort. Finally, we selected the significant SNPs based on the interactions of *p*-values.

### 2.3. Measurement of Anthropometric and Biochemical Parameters

The study participants completed a medical history and lifestyle questionnaire, and then underwent a health examination by trained clinicians according to a standard protocol [17]. The status of smoking was subdivided into three groups (current-, ex-, and non-smokers). The drinking status (alcohol intake) was categorised into three groups (current-, ex-, and non-drinkers). The regularity of physical activity was determined according to whether subjects participated regularly in any sports to the point of sweating. Among the dyslipidaemia subjects, treatments of dyslipidaemia were subdivided into seven groups (drugs, diet, fitness, drugs and diet, drugs and fitness, diet and fitness, and drugs, diet and fitness). The BMI was the weight in kilograms divided by the squared height in meters (kg/m^2^). The systolic blood pressure (SBP) and diastolic blood pressure (DBP) were measured twice using a standardised mercury sphygmomanometer (Baumanometer-Standby; W.A. Baum Co. Inc., New York, NY, USA). Blood samples were collected after overnight fasting in a plain tube. Biochemical parameters, such as the total cholesterol, HDL cholesterol, and triglycerides, were determined using enzymatic methods (ADVIA 1650, Siemens, Tarrytown, NY, USA). The LDL-cholesterol was calculated using the Friedewald equation (LDL = total cholesterol − HDL − (TG/5)).

### 2.4. Definitions of Dyslipdemia and CVDs (CAD, IS, and CCD)

Dyslipidemia was diagnosed by physicians, the current use of lipid-lowering medication, or following the National Cholesterol Education Program—Adult Treatment Panel III (NCEP-ATP III) criteria: (1) hypercholesterolemia (TC ≥ 240 mg/dL), (2) hypo-HDL-cholesterolemia (HDL-C < 40 mg/dL), (3) hyper-LDL-cholesterolemia (LDL-C ≥ 160 mg/dL), or (4) hypertriglyceridemia (TG ≥ 200 mg/dL). If one or more NCEP-ATP III criteria were met, the subjects were considered as dyslipidemia patients. CVDs consisted of CAD, IS, and CCD. We defined the CAD as a participant-reported history of diagnosis or treatment of angina pectoris or myocardial infarction. The IS was defined in the same manner based on the participant-reported history of the diagnosis or treatment of IS. The CCD was defined as the combination of CAD and IS per our study outcome definition.

### 2.5. Definition of Nutrition Intake

To assess the dietary intakes of Korean adults in this study, a semi-quantitative food frequency questionnaire (FFQ) containing 103 items was implemented for the KoGES [19]. The FFQ is one of the useful tools for investigating the associations between diet habits and chronic diseases in large population-based studies. The participants answered with the frequency and amounts of foods eaten over the past year. We set the nutrition intake criteria based on the 2020 Korean Dietary Reference Intakes (DRIs) (Ministry of Health and Welfare’s research project, 2020 KDRIs), the usual term for a set of reference values used to plan and assess the nutrient intakes of people. For macronutrients (such as carbohydrates, proteins, and fats), an acceptable macronutrient distribution range (AMDR), a range of intakes for energy sources, was considered. The AMDRs for carbohydrates, proteins, and fats were 55–65%, 7–20%, and 15–30%, respectively [20,21].

The proportions of nutrient intake were categorised as follows: carbohydrate intake: high ≥ 60%, 60 > moderate ≥ 50%, and 50% > low; protein intake: high ≥ 30%, 30 > moderate ≥ 20%, and 20% > low in daily TEI; and fat intake: high ≥ 40%, 40 > moderate ≥ 30%, and 30% > low [20].

### 2.6. Genotyping and Quality Control Procedures

The genotypes were provided by the Center for Genome Science, Korea National Institute of Health (KNIH). The genotypes were produced by the Korea Biobank Array (KORV 1.0, Affymetrix, Santa Clara, CA, USA) [22]. The experimental results of the array were filtered by following quality control procedures and criteria: call-rate > 97%, minor allele frequency (MAF) > 1%, and Hardy–Weinberg equilibrium test *p*-values > 1 × 10^–5^. After the quality control procedures, the experiment genotypes were phased using ShapeIT v2, and IMPUTE v2 was used for imputation analyses of the genotype data with 1000 Genomes Phase 3 data for the reference panel. After the imputation, the imputed variants of quality score < 0.4 or MAF < 1% were excluded from further analyses [22]. The total number of SNPs for the GWAS was 7,975,321 from chromosomes 1 to 22. We annotated the closest or nearby genes of the highly significant variants as candidate genes using LocusZoom version 0.4.8.2 (http://csg.sph.umich.edu/locuszoom) [23].

### 2.7. Statistical Analysis

All data are presented as an average ± standard deviation (SD) or number and percent. The associations between individual SNP genotypes and CVD risk were assessed using an additive mode analysis for each copy of the minor allele and a logistic regression analysis adjusted by age, sex, BMI, exercise status, smoking status, alcohol intake, and principal component (PC) 1 and PC 2 as covariates. All genetic association tests were conducted using PLINK version 1.9 (https://www.cog-genomics.org/plink) [24]. PC1 and PC2 were obtained via the principal component analysis, which was conducted to reduce the bias of the genomic data because of the regional differences in the sample collection. We selected a significant locus via cluster SNPs, with an SNP gap of less than 50 kb and with high linkage disequlibrium (LD r^2^ > 0.8). The significant associations were defined by genome-wide significance level *p*-values (<5 × 10^–8^) [24]. The gene-region plot of the top SNP associations was plotted using a web-based software (LocusZoom version 0.4.8.2, http://csg.sph.umich.edu/locuszoom) [23].

## 3. Results

The clinical characteristics of the 58,610 participants in the KoGES HEXA are presented in Table 1. A total of 20,596 subjects with dyslipidaemia (58.3% female, age 55.3 ± 7.6 years) were included (Table 2). The lists of genome-wide significant (*p*-value < 5 × 10^−8^) and suggestive (5 × 10^−8^ ≤ *p*-value < 1 × 10^−5^) SNPs from the GWAS are available. The GWAS of TCHL (Appendix A) showed 3772 genome-wide significant SNPs, HDL (Appendix A) showed 5098 significant SNPs, LDL (Appendix A) showed 3653 significant SNPs, and TG (Appendix A) showed 4455 significant SNPs. The GWAS of TCHL (Appendix A) showed 8388 genome-wide suggestive SNPs, HDL (Appendix A) showed 9018 suggestive SNPs, LDL (Appendix A) showed 7557 suggestive SNPs, and TG (Appendix A) showed 6672 suggestive SNPs. The data for the 4 GWASs are illustrated in Figure 3 as Manhattan plots using log10-transformed *p*-values. The most significant SNPs in each chromosome were described as the related genes (Figure 3).

Of the SNPs in Figure 3, we excluded those with a different trend compared with those in the Asian, European, African, Japan GWAS, Ansan, Ansung, and Nongchon cohorts. Therefore, we selected SNPs with the same trend in both KoGES HEXA and the world-wide cohort of KoGES. The selected SNPs associated with lipid traits (TCHL, HDL, LDL, TG) for each group are described in Appendix A.

With the filtered SNPs, we analysed the genetic association between traits and CVDs, as shown in Appendix A. The analysis was conducted in dyslipidaemia patients. The leading SNP in CCD (rs2070895) showed the strongest association (*p* = 1.78 × 10^−2^). The odds ratio was 0.8956, which implies that with the minor allele rs2070895, the prevalence of CCD decreases. The SNP also suggested a β value of 1.97, showing that as the minor allele (A) increases by 1, the value of HDL-cholesterol (mg/dL) increases by 1.97 (*p*-value = 9.2 × 10^−158^) in KoGES HEXA. We then analysed the genetic interaction of nutrient intake and CCD, as shown in Appendix A. After adjusting for age, sex, and nutrient intake for carbohydrates, proteins, and fats (model 1), the interaction *p*-value in the protein diet was 4.28 × 10^−3^. After further adjusting model 1 for BMI, smoking history, alcohol uptake, exercise status, PC1, and PC2 (model 2), the interaction *p*-value in the protein diet was 6.12 × 10^−3^. The SNP rs2070895 showed a genetic interaction between protein intake and the prevalence of CCD in dyslipidaemia patients (Appendix A).

The dyslipidaemia patients were subdivided into three groups according to the proportion of protein intake: low < 20%, normal 20–30%, and high ≥ 30%. Each subgroup was also subdivided into three smaller groups according to genotype (GG, GA, and AA type), and the prevalence of CCD in each group was measured. In the GG type, the prevalence of CCD was 1.32% in the normal protein intake group; however, the prevalence of CCD was 5.84% in the low protein intake group. In the GA type, 3.30% were in the normal protein intake group; however, the prevalence of CCD was 4.54% in the low protein intake group. Thus, as the proportion of protein intake decreased, the prevalence of CCD increased in both the GG and GA types.

## 4. Discussion

This large-scale GWAS demonstrated that there are SNPs associated with dyslipidaemia traits, and certain SNPs have genetic association with CVDs. Furthermore, the SNPs show genetic interactions with nutrient intake and CVDs in the dyslipidaemia group. In our study, rs2070895 was related to HDL-cholesterol and interacted with protein intake and the prevalence of CCD. The SNP with a minor allele is positively associated with the value of HDL-cholesterol. Moreover, the SNPs genetically interacted with protein intake and the prevalence of CCD. Subjects with a low intake of protein had a higher prevalence of CCD than participants with a normal intake of protein in the GG and GA types of SNP.

rs2070895 is located on 15:58723939 and associated with the *LIPC* gene. The *LIPC* gene encodes hepatic lipase, which is synthesised and secreted from the liver [25]. Hepatic lipase is a lipolytic enzyme that plays a role in HDL metabolism by hydrolysing triglycerides and phospholipid in HDL; the hepatic lipase activity is inversely associated with the level of HDL cholesterol [26,27]. The hepatic lipase activity is influenced by *LIPC* variants, indicating the fact that the HDL level can be affected by genetic factors. A previous report showed a significant association of the minor allele in the *LIPC* polymorphism with lower hepatic lipase activity [28]. Therefore, the genotype with a minor allele can increase the level of HDL-cholesterol due to the decrease in hepatic lipase activity. In other words, rs2070895 is associated with HDL by regulating the hepatic lipase activity. Consequently, we focused on the effect of HDL on CCD.

Although it is challenging to reveal the causality of protein intake in CCD, we propose possible mechanisms by considering the function of HDL as a mediator linking protein intake and CCD. HDL-cholesterol exhibits an anticoagulant effect exerted by protein C and protein S. HDL enhances activated protein C (APC) inactivation of factor Va, as well as protein S cofactor activity [29].

Protein C functions as an anticoagulant. It enhances the inactivation of factors Va and VIIIa in the bloodstream [30,31]. Protein C is activated by protein S. Protein C and protein S are, thus, antithrombotic factors. In the coagulation pathway, HDL functions as a cofactor to the APC, which stimulates the degradation of factors Va and VIIIa [32]. Furthermore, HDL increases APC and protein S anticoagulant activity in the normal plasma [29]. In these mechanisms, HDL-cholesterol has antithrombic effects, and consequently many studies have shown that HDL-cholesterol is inversely associated with CAD, atherosclerosis, and IS [33,34,35].

A healthy diet that includes reducing the intake of saturated and trans fatty acids and enhancing the intake of vegetables, fish, and fruit can increase HDL-cholesterol [36]. Typically, such intake is achieved with a Mediterranean diet with a carbohydrate/protein/fat ratio of 40%:30%:30% [37]. A high-protein diet can increase HDL-cholesterol in dyslipidaemia patients [38]. A high-protein diet can also decrease the risk for CVDs through the increase in HDL-cholesterol [39].

Despite this study being a large-scale GWAS, there are some limitations. First, we analysed the association of diet with the prevalence, not incidence, of CCD. Therefore, we cannot conclude causality. Second, we defined CVDs depending on a participant-reported questionnaire and might have missed patients who did not know they had a CVD. Third, because KoGES data had provided the history of dyslipidaemia drugs (yes or no) based on the self-reported questionnaire, specific types of the drugs could not be described in our study. Fourth, these results cannot demonstrate whether the joint effect of the genotype and protein diet has synergistic effects on CCD. Fifth, the participants enrolled in the study were Koreans. In order to be a representative of the world, further studies analysing the world-wide GWAS data are needed. Furthermore, there have been few subjects with a high protein intake, and it is difficult to compare a high-protein group with normal- or low-protein group. Finally, this research was based on a statistical analysis and did not allow for experimental validation. Genotype interactions assessed by conducting biological and mechanical analyses are needed to reinforce these findings, and additional studies are required for validation. Moreover, further prospective studies associated with the style of diet, such as the Mediterranean-style diet or DASH diet, are needed.

Despite these limitations, our study has several strengths. This study used representative national data and included a large number of subjects (approximately 60,000). Previous studies have found various genetic relationships between CVDs and its well-known risk factors, based only on pre-selected SNPs [40,41,42]. Our study demonstrated a novel genetic interaction with CVDs and nutrient intake based on newly selected SNPs by comparing GWAS in KoGES HEXA with a world-wide GWAS.

## 5. Conclusions

Our GWAS with an unprecedented study design provides new insights into the genetic architecture of CVDs. We found that as the proportion of protein decreases, the prevalence of CCD increases in dyslipidaemia patients with the GG/GA type of the *LIPC* gene. Therefore, sufficient protein uptake is important to prevent CCD. The prevalence of CCD was higher in subjects with the GG type than those with the GA type when the proportion of protein intake decreased because the G type is more vulnerable to dyslipidaemia.

## Figures and Tables

**Figure 1 biomedicines-10-01615-f001:**
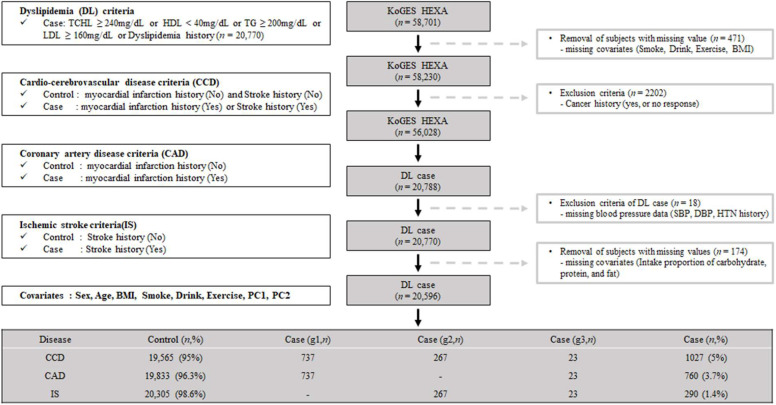
Flow chart of study population selection. g1, the patients group only with Coronary Artery Disease (CAD); g2, the patients group only with Ischemic stroke (IS); g3, the patients group with both Coronary Artery Disease (CAD) and Ischemic stroke (IS).

**Figure 2 biomedicines-10-01615-f002:**
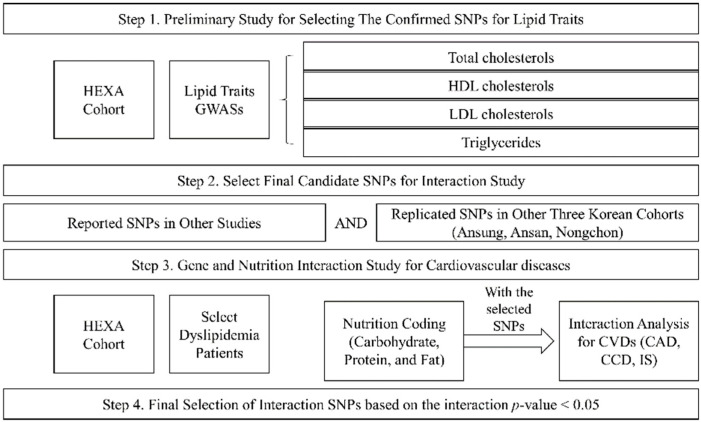
Study design.

**Figure 3 biomedicines-10-01615-f003:**
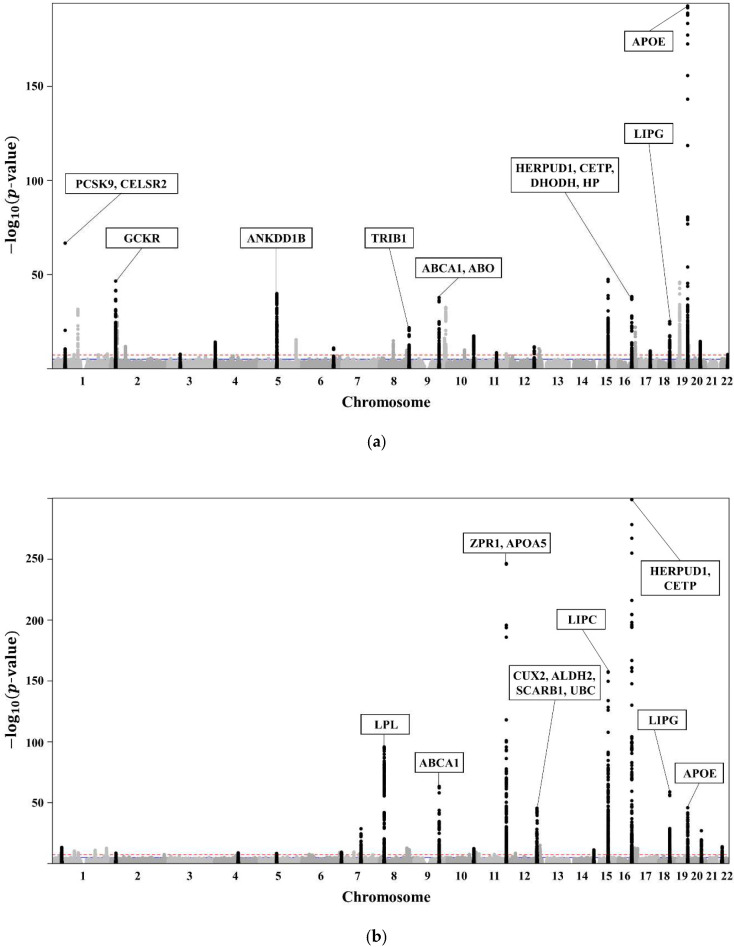
(**a**) Manhattan plot showing the relationships between TCHL and SNPs. (**b**) Manhattan plot showing the relationships between HDL and SNPs. (**c**) Manhattan plot showing the relationships between LDL and SNPs. (**d**) Manhattan plot showing the relationships between TG and SNPs.

**Table 1 biomedicines-10-01615-t001:** Clinical characteristics of the city cohort (KoGES HEXA).

Characteristics	City Cohort (KoGES HEXA)
Number of subjects	58,610
Age, years	53.8 ± 8.0
Female, *n* (%)	38,330 (65.4)
*Anthropometric traits*	
BMI, kg/m^2^	23.89 ± 2.88
*Biochemical traits*	
Total cholesterol, mg/dL	197.15 ± 35.31
HDL cholesterol, mg/dL	53.66 ± 12.95
LDL cholesterol, mg/dL	117.29 ± 35.19
Triglycerides, mg/dL	120.90 ± 69.01
*Lifestyle factors*	
Smoking status: Never/Quit/ Current, *n* (%)	42,765 (73.0)/9243 (15.8)/6408 (10.9)
Drinking status: Never/Quit/ Current, *n* (%)	30,273 (51.7)/2204 (3.8)/25,895 (44.2)
Exercise status: No/Yes, *n* (%)	26,488 (45.2)/31,922 (54.5)
CAD, *n* (%)	1669 (2.9)
IS, *n* (%)	708 (1.2)
CCD, *n* (%)	2326 (4.0)
Total energy, kcal/day	1744.15 ± 551.17
Carbohydrates (%)	71.72 ± 6.98
Protein (%)	13.41 ± 2.57
Fat (%)	13.87 ± 5.40

**Table 2 biomedicines-10-01615-t002:** Clinical characteristics of the dyslipidaemia group in KoGES HEXA.

Characteristics	Dyslipidaemia Group
Number of subjects	20,596
Age, years	55.3 ± 7.6
Female, *n* (%)	12,014 (58.3)
*Treatments*	
Drugs, *n* (%)	2293 (11.13)
Diet, *n* (%)	21 (0.1)
Fitness, *n* (%)	30 (0.15)
Drugs and Diet, *n* (%)	32 (0.16)
Drugs and Fitness, *n* (%)	148 (0.72)
Diet and Fitness, *n* (%)	64 (0.31)
Drugs, Diet and Fitness, *n* (%)	206 (1.00)
*Anthropometric traits*	
BMI, kg/m^2^	24.67 ± 2.82
*Biochemical traits*	
Total cholesterol, mg/dL	212.21 ± 43.91
HDL cholesterol, mg/dL	48.43 ± 13.78
LDL cholesterol, mg/dL	128.51 ± 42.29
Triglycerides, mg/dL	176.37 ± 114.19
*Lifestyle factors*	
Smoking status: Never/Quit/ Current, *n* (%)	13,760 (66.8)/3828 (18.6)/3008 (14.6)
Drinking status: Never/Quit/ Current, *n* (%)	10,501 (51)/882 (4.3)/9213 (44.7)
Exercise status: No/Yes, *n* (%)	9950 (48.3)/10,646 (51.7)
CAD, *n* (%)	760 (3.7)
IS, *n* (%)	290 (1.4)
CCD, *n* (%)	1027 (5)
Total energy, kcal/day	1745.32 ± 548.61
Carbohydrates (%)	72.01 ± 6.96
Protein (%)	13.35 ± 2.58
Fat (%)	13.59 ± 5.38

## Data Availability

Data in this study were from the Korean Genome and Epidemiology Study (KoGES; 4851-302). These data are available through an online sharing service under the permission of the division of epidemiology and health index in the Korean Center for Disease Control and Prevention (KCDC), at http://www.nih.go.kr/NIH/eng/contents/ (accessed on 20 April 2022).

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
