# Peer review of "Dyslipidaemia—Genotype Interactions with Nutrient Intake and Cerebro-Cardiovascular Disease"

_biomedicines, 2022, doi:10.3390/biomedicines10071615_

Round 1

Reviewer 1 Report

The authors investigated the role of genetic variants in dyslipidaemia and their interactions with macronutrients for CV disease using large-scale genome-wide association of Korean adults. Overall the findings suggest a correlation between specific polymorphisms associated with dyslipidaemia and protein intake. Overall the study is well organized and written. 

Before publication, some major issues have to be addressed:

1) The introduction is missing the main clinical consequences of dyslipidaemia.

2) Patient treatment information is missing (big bias, in sub group formation).

2)Limitations are missing (ex: the Korean population is too reductive, multi center centered studies are required)

3)Future prospective are missing

4)What are the clinical treatment implications of this study

Reviewer 2 Report

The presented paper is well written and interesting. However, I am much concerned about matching numbers in abstract and methods. Abstract: 20,596 subjects with dyslipidaemia and 1027 with CCD. While flow chart somehow misses these numbers: there is DL: 20,770, then below Controls with various numbers (19,565; 19829; and 20,301). While in study group cad + stroke is 1050, not 1027. I assume that some patients had both stroke and myocardial infarction, but this should be explained below the chart. Please simplify and clarify Flow chart, as now as it is, it is confusing.

The other issue is that discussion could be more developed. The study is complex, contains large number of patients, with broad attention paid to cardiovascular risk factors, nutrient habits, including proteins intake, and finally genotyping. The Authors could address why HDL level is so important, what is the mechanism of proteins/HDL protection against developing cardiovascular disease.

Round 2

Reviewer 1 Report

The authors have made the required adjustments. The article is now fit for publication.

Reviewer 2 Report

The Authors answered all comments in satisfactory way. 

Minor remarks.

Table 1 is duplicated. The Authors probably ment Table 1 for cohort characteristics. Table 2 for dyspilidemia.

Insteed of writting 'drugs', write statins, or ezetimide, or others in Table 2. 
